# Mitochondrial genome in sporadic breast cancer: A case control study and a proteomic analysis in a Sinhalese cohort from Sri Lanka

**Lakshika P. Jayasekera[1], Ruwandi Ranasinghe[1]\*, Kanishka S. Senathilake[1], Joanne T. Kotelawala[1], Kanishka de Silva[2], Priyanka H. Abeygunasekara[2], Renuka Goonesinghe[2], Kamani H. Tennekoon[1]\***

**1** Institute of Biochemistry, Molecular Biology and Biotechnology, University of Colombo, Colombo, Sri Lanka, **2** National Cancer Institute, Apeksha Hospital, Maharagama, Sri Lanka

\* kamani@ibmbb.cmb.ac.lk (KHT); ruwa@ibmbb.cmb.ac.lk (RR)

**Data Availability Statement:** All relevant data are within the paper and its Supporting information files.

## Abstract

Breast cancer is the commonest malignancy in women and the majority occurs sporadically with no hereditary predisposition. However, sporadic breast cancer has been studied less intensively than the hereditary form and to date hardly any predictive biomarkers exist for the former. Furthermore, although mitochondrial DNA variants have been reported to be associated with breast cancer, findings have been inconsistent across populations. Thus we carried out a case control study on sporadic breast cancer patients and healthy controls of Sinhalese ethnicity (N = 60 matched pairs) in order to characterize coding region variants associated with the disease and to identify any potential biomarkers. Mitochondrial genome was fully sequenced in 30 pairs and selected regions were sequenced in the remaining 30 pairs. Several *in-silico* tools were used to assess functional significance of the variants observed. A number of variants were identified among the patients and the controls. Missense variants identified were either polymorphisms or rare variants. Their prevalence did not significantly differ between patients and the healthy controls (matched for age, body mass index and menopausal status). *MT-CYB*, *MT-ATP6* and *MT-ND2* genes showed a higher mutation rate. A higher proportion of pre-menopausal patients carried missense and pathogenic variants. Unique combinations of missense variants were seen within genes and these occurred mostly in *MT-ATP6* and *MT-CYB* genes. Such unique combinations that occurred exclusively among the patients were common in obese patients. Mitochondrial DNA variants may have a role in breast carcinogenesis in obesity and pre-menopause. Molecular dynamic simulations suggested the mutants, G78S in *MT-CO3* gene and T146A in *MT-ATP6* gene are likely to be more stable than their wild type counterparts.

## Introduction

Breast cancer is the most prevalent malignancy in women world-wide [1] as well as in Sri Lanka [2]. While the incidence of female breast cancer has decreased in some developed countries over the recent years, in Sri Lanka an opposite trend is observed with a ~4% increase per

**Funding:** This work was financially supported by the National Research Council (NRC), Sri Lanka (Grant No: NRC/17-020). The funders had no role in the study design, data collection and analysis, decision to publish or preparation of the manuscript.

**Competing interests:** The authors have declared that no competing interests exist.

year [2, 3]. Majority of breast cancers are sporadic with only 5 to 10% having a hereditary aetiology conferred by mutations in the predisposing genes [4]. Hereditary breast cancer has been extensively studied; genetic biomarkers for its prediction such as mutation screening for *BRCA1* and *BRCA2* are widely used [5]. In contrast there are hardly any predictive biomarkers for sporadic breast cancer despite this being the more prevalent form. Thus, there is an unmet need for predictive biomarkers for sporadic breast cancer, which would enable surveillance of at risk individuals and early detection, thus reducing associated morbidity and mortality.

Mitochondria, the major source of reactive oxygen species are inevitably linked to oxidative stress of the cell. Dysfunction of mitochondria is implicated in carcinogenesis including that of the breast [6, 7]. Constant exposure to reactive oxygen species and use of error-prone pathway for its replication results in a 10 to 100 fold higher mutation accumulation rate in the mitochondrial (mt) genome when compared to the nuclear genome [8]. Both germline and somatic mtDNA variants have been linked to breast cancer in several populations. Such variants have been identified in the control region, a mutation hot spot and in the coding region of mtDNA in breast cancer [9, 10]. However, these findings have remained inconsistent with variants reported to be associated with breast cancer in certain populations showing no association in yet other populations [10–14]. Further, mtDNA variants have been linked to metastasis and treatment resistance in breast cancer [10, 15].

Despite previous research on familial and sporadic breast cancer and related nuclear genes [16–20], only a single study has so far looked into the association of mtDNA and breast cancer in Sri Lankans [21]. Sri Lankan breast cancer patients of Sinhalese ethnicity did not show a significant association with control region variants or mtDNA haplogroups. The present investigation was therefore aimed to identify possible germline variants associated with breast cancer in the rest of the mt genome given the importance of mt genes in the oxidative phosphorylation.

Since mitochondrial genes encode subunits of large protein complexes and is highly polymorphic, the pathogenic nature of a newly identified rare sequence variation cannot be established solely by the frequency of occurrence in a limited population. Although functional studies are the gold-standard to determine the pathogenicity, it is not practical or even possible to perform *in vitro* or *in vivo* experiments for each novel variant. However the pathogenic nature of the variations may be initially predicted using a series of criteria before carrying out functional studies *in vitro* or *in vivo* [22, 23]. Accordingly, several bioinformatics tools have been developed to predict the pathogenicity of DNA variants.

The present investigation was aimed to identify coding region germline variants of the mt genome associated with breast cancer and to ascertain possible pathogenicity by predicting their functional consequences using bioinformatics software. Using a matched-pairs study design, the coding region of the mitochondrial genome was compared between sporadic breast cancer patients and healthy controls matched for age, body mass index and menopausal status. Prevalence of identified variants were compared between the two groups and subsets based on the menopausal status, histology of the tumour and body size. The rate of mutation was computed for each mitochondrial gene. Variants identified were tested *in-silico* for functional significance using several tools and selected variants were further analysed using Molecular Dynamic simulations. S1 Fig gives the summary of the study protocol.

## Materials and methods

### Study population

Ethical approval for the study was granted by the Ethics Review Committee of the Faculty of Medicine, University of Colombo (EC-16-222). All the study participants provided written

informed consent before admission to the study. Primary breast cancer patients (N = 60) of Sinhalese ethnicity with histologically confirmed diagnosis of breast cancer were recruited from the National Cancer Institute, Sri Lanka between December 2016 and February 2019. None of them had a family history of breast or any other cancer. Healthy Sinhalese women (n = 60) matched for age (±5 years), body mass index (±1) and menopausal status, with no personal or family history of any cancer were recruited from the community as the control group. Forty pairs were newly recruited. The remaining 20 pairs were selected from a previous study which used same inclusion and exclusion criteria [21]. Peripheral venous blood samples were collected from the patients before commencing chemotherapy, radiotherapy or neoadjuvant therapy and from the controls at the time of admission to the study. Thirty of the newly recruited pairs were subjected to whole mt genome sequencing using a next generation sequencing (NGS) platform to identify mtDNA variants. The remaining 30 pairs were analyzed for the genomic regions *MT-ND3*, *MT-ND4L* and *MT-CYB* using Sanger sequencing and the primer sets used covered parts of *MT-ND6* and *MT-TT* as well.

## DNA extraction and quantification

Total genomic DNA was extracted from the newly recruited 40 pairs using QIAmp DNA Blood Mini Kit (Catalogue no: ID: 51104 Qiagen, Hilden, Germany). DNA had been extracted using modified Miller's salting out procedure from the remaining 20 pairs [24]. The quality and quantity of extracted DNA were assessed using BioSpec Nano spectrophotometer (Model no: A115746, Shimadzu, Tokyo, Japan).

## Next generation sequencing of the whole mtDNA and bioinformatics analysis

Whole mt genome of 30 newly recruited pairs was sequenced at a commercial NGS service facility (Genotypic Technology, Bangalore, India). Mitochondrial amplicons were initially generated through PCR to cover the complete human mitochondrial genome. The amplicons were then pooled, cleaned up and subjected to Illumina compatible library preparation by fragmentation and adaptor ligation. The fragmented DNA was further subjected to indexing and enrichment through another round of PCR, followed by purification, quality check and Illumina sequencing (Hiseq/ Nextseq 500 systems). The raw paired-end sequencing data received in fastq format from the service provider were then analyzed in house.

The quality of the reads was assessed with Fastqc (version 0.11.8) [25]. Adapters, short reads and reads below accepted quality were removed using cutadapt tool (version 1.18) [26]. Reads were aligned against indexed revised Cambridge Reference Sequence (rCRS; GenBank accession number NC_012920.01) [27] with BWA-MEM aligner (version 0.7.12-r1039) [28]. Using SAMtools (version 1.9), the resulting sequence alignment mapping (SAM) format file was then converted to a binary alignment map (BAM) [29, 30]. Same tool was used to sort and index the BAM file and duplicates were removed. Base quality recalibration was done with the tools base quality score recalibration (BQSR), BaseRecalibrator and ApplyBQSR by Genome Analysis Tool Kit (GATK) by Broad Institute of Harvard and MIT (version 4.2.0.0) [31] upon preprocessing the input files with Picard (version 2.25.1). GATK AnalyzeCovariates tool was additionally used to evaluate the effects of recalibration. The variants were called and filtered with the tools of GATK (version 4.2.0.0), Mutect2—mitochondria mode [32] and FilterMutectCalls respectively. The variants that passed the applied filters were selected with SelectVariants tool of the same tool kit to produce the final VCF output. Mutant allele fraction >0.90 was considered homoplasmic and 0.1 to 0.90 as heteroplasmic. The variants were visually examined and confirmed with Integrative Genomics Viewer (IGV version 2.9.2) [33] followed

by annotation with Variant Effect Predictor (VEP) [34] and further reviewed in MITOMAP data base [35]. Locally installed Haplogrep 2 (version 2.1.25) [36] and PhyloTree build 17 [37] were used to assign haplogroups of the final output files.

## Sanger sequencing

DNA from the remaining 30 matched pairs were subjected to Sanger sequencing to sequence genomic regions of *MT-ND3*, *MT-ND4L* and *MT-CYB*. Selected regions were amplified using 3 sets of primers previously described [38]. The primer pair 15 (forward, 5'-TCT CCA TCT ATT GAT GAG GGT CT-3'; reverse, 5'-AAT TAG GCT GTG GGT GGT TG-3'), spanning ~891 bp region of human mtDNA from 9989 to 10837 was used to amplify the region of *MT-ND3* (10059–10404) and *MT-ND4L* (10470–10766) genes. The overlapping primer pairs 21 (forward, 5'- GCA TAA TTA AAC TTT ACT TC -3'; reverse, 5'- AGA ATA T TG AGG CGC CAT TG -3') , and 22 (forward, 5'- TGA AAC TTC GGC TCA CTC CT -3'; reverse, 5'- AGC TTT GGG TGC TAA TGG TG -3') spanning ~1978 bp region of human mitochondrial DNA across positions 14000–15978 were used to amplify the complete *MT-CYB* gene (14747–15887) and the amplicon included a part of the *MT-ND6* (14200–14673 region) and a part of the *MT-TT* (15888–15930 region) genes. The optimized reaction mixture adjusted to a final volume of 25 μL contained 50 to 200 ng genomic DNA, 0.2 μM (final concentration) of each primer (Integrated DNA technologies, Coralville, Iowa, United States) and 12.5 μL of EmeraldAmp® GT PCR 2X premix of the master mix (Takara Bio-Inc, Kusatsu, Shiga, Japan). The PCR reactions were carried out in Veriti Thermal Cycler (Catalogue no: 4375786, Thermo Fisher Scientific, Inc., Waltham, Massachusetts, United States) with following thermal cycling program—initial denaturation at 94°C for 1 min, 35 cycles of denaturation at 94°C for 30 s, primer annealing at 58°C for 45 s, primer extension at 72°C for 2 min and final extension completed at 72°C for 5 min. PCR products were then purified using Wizard® SV Gel and PCR Clean-Up System columns (Catalogue no: A9281, Promega Corporation, Madison, Wisconsin, United States). The Sanger sequencing was carried out with BigDye® Terminator v3.1 cycle sequencing kit (Catalogue no: 4337455, Thermo Fisher Scientific, Inc. Waltham, Massachusetts, United States), using the same pairs of primers used for the thermal programme. The sequencing products were then subjected to ethanol precipitation, evaporated to dryness and resuspended in 12 to 15 μ of Highly-Deionized formamide (Catalogue no: 4401457, Thermo Fisher Scientific, Inc.) followed by heating for 5 min at 95°C. Samples were then loaded on to Applied Biosystems 3500Dx Genetic Analyzer (Catalogue no: 4337455, Thermo Fisher Scientific, Inc). Sequences obtained were analyzed against the rCRS with Bio Edit sequence alignment editor software (version 7.2.5) and Variant Surveyor® (version 4.09).

## Statistical analysis

McNemar's test was performed (GraphPad QuickCalcs: McNemar test online https://www.graphpad.com/quickcalcs/McNemar1.cfm, GraphPad Software, San Diego, California, USA) to identify variants that are significantly associated with breast cancer. P<0.05 was considered statistically significant. Data from the whole mt genome analysis and Sanger sequencing were pooled for the statistical analysis. As one histological type of breast cancer accounted for 45 matched pairs, we further analysed this group separately. Proportions of pre-menopausal patients and postmenopausal patients having missense or pathogenic variants and, proportion of obese and non-obese patients having exclusive combinations of missense variants were compared using Z score for two population proportions (https://www.socscistatistics.com/tests/ztest/).

## Bioinformatic analysis of the coding region variants identified

Non-synonymous variants identified through both NGS and Sanger sequencing were searched in MITOMAP [35] database to identify previously reported variations and their association with disease phenotypes. Variants with no confirmed data on the association with breast cancer or any other disease phenotype (either through functional studies or based on frequency) were further analyzed using bioinformatics tools to detect possible functional impact of variations.

First, the evolutionary conservation for each SNP was visualized by multiple alignment of sequences from different species using ConSurf server [39] and Clustal Omega [40]. Whenever available, functional annotation data (i.e. reported functional residues, pattern and regions) were also obtained through InterPro scan and from literature to predict the functional importance of the residues at SNPs. SNAP2 server [41], Hmtvar [42] and MitImpact 3D [43] Databases, SIFT and Polyphen2 servers as well as APOGEE and CADD Databases within MitImpact 3D were used to further analyze the pathogenicity and functional impact of non-synonymous variants based on different criteria.

## Molecular dynamics simulation

Variations present in the patients at highly conserved regions (ConSurf score $\geq 6$) and predicted to be deleterious/ pathogenic in nature by at least four out of five prediction servers were further evaluated for their functional consequences by molecular dynamics (MD) simulations.

The MD simulations studies were carried out using wild type human cytochrome C oxidase (PDB ID: 5Z62) and its mutant G78S in mitochondrial membrane using the Desmond 2020.1 [44]. The human mitochondrial ATPase (wild type) and its mutant T146A were modelled using Alphafold2 (https://www.alphafold.ebi.ac.uk/) and subjected to MD simulation studies. Mutant G78S in Chain A was generated using UCSF Chimera by selecting specific residues Glycine78 and altered with Serine and followed by modelling. The lipid bilayer membrane (POPE) was developed using the membrane builder module in Schrodinger Maestro. The OPLS-2005 force field [45–47] and explicit solvent model with the TIP4P water model were used in this system [48] in period boundary salvation box of 10 Å x 10 Å x 10 Å dimensions. $Na^+$ ions were added to neutralize the charge 0.15 M, NaCl solutions were added to the system to simulate the physiological environment. Initially, the system was equilibrated using an NVT ensemble (Canonical ensemble; N—number of molecules, V- volume, T–temperature) for 10 ns to retrain over the wild type and the mutant in the membrane. Following the previous step, a short run of equilibration and minimization was carried out using an NPT ensemble (Iso-thermal-isobaric ensemble; N—number of molecules, P–pressure, T–temperature) for 12 ns. The NPT ensemble was set up using the Nose-Hoover chain coupling scheme [49] with the temperature at 37˚C, the relaxation time of 1.0 ps, and pressure 1 bar maintained in all the simulations. A time step of 2fs was used. The Martyna-Tuckerman–Klein chain coupling scheme [50] barostat method was used for pressure control with a relaxation time of 2 ps. The particle mesh Ewald method [51] was used for calculating long-range electrostatic interactions, and the radius for the coulomb interactions were fixed at 9Å. RESPA integrator was used for a time step of 2 fs for each trajectory to calculate the bonded forces. The root means square deviation (RMSD), radius of gyration (Rg) and the number of hydrogen (H-bonds) were calculated to monitor the stability of the MD simulations.

## Principal Component analysis (PCA)

During a 20 ns simulation of the complex wild type human cytochrome C oxidase and its mutant G78S and wild type mitochondrial ATPase and its mutant T146A, PCA analysis was

used to recover the global movements of the trajectories. A covariance matrix was generated as described to calculate the PCA. Conformational analysis of wild type and mutant proteins were done by using 2 appropriate modes of conformation. The main component was calculated as trajectories, and a comparison of the first highest mode (PC1) versus the second highest mode (PC2), was investigated. Geo measures v 0.8 were used to calculate the principal components based on eigenvectors from the correlation matrix on both the proteins. The MD trajectory versus principal components of trajectory motion was recorded in a 2D plot using the Matplotlib python package using Geo measures, including a comprehensive library of g sham [52].

## Results

The study population of 60 matched pairs of sporadic breast cancer and healthy controls comprised of women aged between 30 to 74 years (mean±SD: patients 50.22±10.16 years; controls 49.80±9.54 years). Majority (60% of the pairs) were pre-menopausal. Over 70% of the study group was either obese or overweight (S1 Table). Invasive ductal carcinoma [i.e.: not otherwise specified (NOS) and carcinoma of no special type (NST) [53, 54] accounted for 75% of the cancers (S2 Table).

### MtDNA variants

The mean±SD of genome wide depth coverage (post removal of duplicates) acquired for the 60 samples analysed using NGS (patients = 30, matched controls = 30) was 251.9±21.283 (S2 Fig). A total of 503 variants were identified, 464 in the whole mt genome through NGS and remaining 39 through Sanger sequencing. Of the variants identified through NGS, 26 distinct positions were present in heteroplasmy. Of the 503 variants identified, 138, 186 and 179 were respectively found only among the patients, only among the controls or in both groups. Majority (487) were base substitutions. Seven deletions and 9 insertions were found. Out of the 30 pairs subjected to NGS, all except 5 patients (16.66%) and 8 healthy controls (20.00%), showed heteroplasmy at one or more positions along the mitochondrial genome.

All but eight variants (248delA, 310insC, 514_515delCA, 655insC, 8270insACCCCCTCT, 16162delA, 16180delA, 16180_16181delAA) identified in the present study were reported in Mitomap. Description of the variants found at a frequency >5% across the mitochondrial coding region are given in S3 and S4 Tables. Altogether 86 missense variants were observed, but only 25 of these occurred at a prevalence of >5% in the study population. Highest number of missense variants were seen in the *MT-ND2* gene followed by *MT-ATP 6*, *MT-CYB* and *MT-ND5* genes. Some missense variants were seen only in the patients, others only in the controls. The remainder occurred in both the patients and controls, but the prevalence did not significantly differ between the two groups. More than 50% of the patients and controls carried the variants, A8701G and A8860G in *MT-ATP6*, A10398G in *MT-ND3*, C14766T and A15326G in *MT-CYB*. Of these, A8860G variant in *MT-ATP6* and the two variants in *MT-CYB* occurred in 90% to 100% of the patients and the controls studied. Interestingly several variants were observed exclusively in patients with invasive ductal carcinoma (NST/NOS) and these included a missense variant each in *MT-ND1* (A3434G), *MT-ND2* (G4491A), *MT-TD* (T7581C), *MT-CO2* (G7859A), *MT-ND3* (G10143A) (S5 Table). Furthermore, some variants observed among other histological types of cancer were not seen in this group.

Six variants (two each in *MT-ND2*, *MT-ATP-6* and *MT-ND4*) were predicted to be deleterious by SIFT as well as probably/possibly damaging by PolyPhen2 (Table 1 and S5 Table). These were found only in 3 patients but in 7 controls. One variant in *MT-ND2* that was predicted to be deleterious by SIFT but benign by PolyPhen2 occurred in one control. There were

**Table 1. Missense variants predicted to be deleterious by _in silico_ analyses.** Only the variants with SNAP2 score >50 or ConSurf conservations score >8 are shown. (see S5 Table for complete list of missense variations) a B: Benign, D: deleterious, lc: low confidence, N: neutral; P: pathogenic; PRD: probably damaging, PSD: possibly damaging, T: tolerated, U: unknown. Variants subjected to molecular dynamic simulations are given in bold.

| GENE | Patients (%) | Controls (%) | Variation | | SNAP2 | SIFT | Poly-phen2 | APOGEE | CADD | Con Surf |
|------|------|------|------|------|------|------|------|------|------|------|
| ND1 | 0.00 | 3.33 | T3394C | Y30H | 87 | D(lc) | B | P | N | 4 |
| | 3.33 | 0.00 | A3397G | M31V | 57 | D(lc) | B | P | N | 7 |
| ND2 | 0.00 | 3.33 | A4638G | I57V | -9 | T | B | N | N | 8 |
| | 0.00 | 6.66 | C4640A | I57M | 53 | D | PSD | P | D | 8 |
| | 0.00 | 3.33 | C5461T | A331V | 67 | T | B | N | N | 1 |
| ATP6 | 0.00 | 6.66 | G8572A | G16S | 78 | D | PSD | N | D | 2 |
| | 6.66 | 6.66 | C8684T | T53I | 54 | T | B | N | N | 6 |
| | 3.33 | 0.00 | T8705C | M60T | 65 | T | B | P | N | 7 |
| | 0.00 | 3.33 | A8812G | T96A | 40 | T | PRD | N | D | 9 |
| | 100.0 | 96.6 | A8860G | T112A | 66 | T | B | N | N | 7 |
| | **3.33** | **0.00** | **A8962G** | **T146A** | **56** | **D** | **PRD** | **N** | **D** | **8** |
| CO3 | **3.33** | **0.00** | **G9438A** | **G78S** | **83** | **D(lc)** | **B** | **P** | **D** | **6** |
| ND4 | 0.00 | 3.33 | G11453A | A232T | 14 | D(lc) | PRD | P | D | 9 |
| ND5 | 3.33 | 0.00 | A13105G | I257V | -35 | T(lc) | B | N | N | 9 |
| | 3.33 | 3.33 | G13708A | A458T | 62 | T(lc) | B | P | N | 8 |
| CYB | 3.33 | 5 | G15110A | A122T | 55 | T(lc) | B | N | N | 1 |
| | 93.33 | 98.33 | A15326G | T194A | 55 | T(lc) | B | N | N | 1 |
| | 10 | 8.33 | G15431A | A229T | 62 | D(lc) | B | P | D | 1 |
| | 3.33 | 0.00 | G15497A | G251S | 81 | N | B | N | D | 4 |

several variants predicted to be tolerated, tolerated with low confidence or deleterious with low confidence by SIFT but predicted to be possibly or probably damaging by PolyPhen2. These occurred in the _MT-ND2_, _MT-ATP8_, _MT-ATP6_, _MT-ND4_, _MT-ND5_ and _MT-CYB_ genes and were seen either in the patients, controls or in both groups. A number of other variants observed had been reported as deleterious and/or pathogenic in CADD and APOGEE Databases (Table 1).

SNAP2 server attributes a score reflecting the possibility of each mutation to change the wild-type protein function, with score > 50, -50 to <50 and <-50 being considered as indicative of strong, weak and no or neutral effect. Accordingly, 15 variants (5 in _MT-ATP6_, 4 in _MT-CYB_, 2 each in _MT-ND1_ and _MT-ND2_, and 1 each in _MT-CO3_ and _MT-ND5_) gave scores >50 (Table 1). Five of these were present only in the patients, 4 only in the controls and the remaining 6 were common to both groups. The highest score of 87 was attributed to the T3394C variant in the _MT-ND1_ gene detected only in 1 control individual. The SIFT server categorized this variant as deleterious with low confidence and APOGEE server indicated pathogenicity. The second highest score of 83 was for the G9438A variant in the _MT-CO3_ gene detected in a single patient. This variant was also categorized as deleterious with low confidence by SIFT while both APOGEE and CADD servers predicted pathogenic nature. The third highest predicted score of 81 was for the G15497A variant in the _MT-CYB_ gene present only in a single patient and only the CADD server predicted it to be deleterious.

Prevalence of missense variants and pathogenic variants that occurred exclusively in patients were more frequent among pre-menopausal patients than among postmenopausal patients. Only the missense variants showed a significant difference in a one tail Z test for two population proportions (58.33% vs 33.33%: P = 0.0287 one tail, P = 0.0576 two tail).

Interestingly several combinations of missense variants occurred in a given gene in some patients and controls (Table 2). Majority of the unique combinations of missense variants

**Table 2. Unique combinations of co-existing missense variants and pathogenicity prediction by SIFT and Polyphen2 and reported in CADD and APOGEE.** Variants for which SNAP 2 score was >50 or con-surf value was 8 or more are shown in bold.

| Co-existing Variants | | | | | | | | | | | | Pat (N) | Con (N) |
|---|---|---|---|---|---|---|---|---|---|---|---|---|---|
| *MT-ATP6 gene* | | | | | | | | | | | | 22 | 25 |
| **G8572A** | G8584A | T8618C | **C8684T** | A8701G | **T8705C** | T8843C | **A8812G** | A8860G | **A8962G** | G9064A | C9094T (T/PSD/D/N) | | |
| **(D/PSD/D/N)** | (T/B/N/N) | (T/B/N/N) | **(T/B/N/N)** | (T/B) | **(T/B/N/N)** | (T/B/N/N) | **(T/PRD)** | (T/B/N/N) | **(D/PRD/D/N)** | (T/B/N/N) | | | |
| | | | √ | | | | | √ | | | | 13 | 18 |
| | | | √ | | | | | √ | | | | 2 | 2 |
| | | √ | | | | | | √ | | | | 1 | 1 |
| | | | | | | √ | √ | | | | | 0 | 1 |
| | √ | | | | | | | √ | | | | 1 | 0 |
| √ | | | | | | | | √ | | | | 0 | 1 |
| | | | | | | | | √ | | | √ | 1 | 2 |
| | | | | √ | | | | √ | √ | | | 1 | 0 |
| | | | | √ | | | | √ | | √ | | 1 | 0 |
| | | | | √ | √ | | | √ | | | | 1 | 0 |
| | | | | √ | | √ | | √ | | | | 1 | 0 |
| *MT-CYB gene* | | | | | | | | | | | | 52 | 57 |
| C14766T | G14861 | A14927 | **G15110** | G15119 | G15314 | **A15326** | **G15431** | C15452 | **G15497** | G15803 | G15884A (T-lc /PSD/D/P) | | |
| (D-lc/B) | A | G | **A** | A | A | **G** | **A** | A | **A** | A | | | |
| | (T-lc/B/ | (D-lc/B/ | **(T-lc/ | (D-lc/B/ | (T-lc/B/ | **(T-lc/B/ | **(D-lc/B/ | (B/N/ | **(N/B/ | (T-lc/B/ | | | |
| | D/N) | D/N) | B/ /N/N)** | D/P) | D/P) | N/N)** | D/P)** | N/N) | D/N)** | N/P) | | | |
| √ | | | | | | √ | | | | | | 37 | 41 |
| | √ | | | | | √ | | | | | | 1 | 0 |
| √ | | | | | | √ | √ | | | | | 5 | 3 |
| √ | | | | | | √ | | | √ | | | 1 | 0 |
| √ | | | | √ | | √ | | | | | | 1 | 1 |
| √ | | | √ | | | √ | | | | | | 2 | 4 |
| √ | | | √ | | | √ | | | | | | 1 | 0 |
| √ | | √ | | | | √ | | | | | | 0 | 1 |
| √ | | | | | | √ | | | | | √ | 2 | 3 |
| √ | | | | | √ | √ | √ | | | | | 0 | 2 |
| √ | | | | | | √ | | √ | | | | 1 | 2 |
| √ | | | | | | √ | | | √ | | | 1 | 0 |
| √ | | | | | √ | √ | | | | | √ | 1 | 1 |
| | | | | | | | | | | | | 0 | 3 |
| *MT-ND2 gene* | | | | | | | | | | | | 2 | 2 |
| **C4640A (D/PSD/D/P)** | C4654T | | A5301G | | A5319G | | **C5461T** | | T5503C | | | | |
| | (T/B/D/N) | | (T/B/N/N) | | (D/B/D/N) | | **(T/B/N/N)** | | (T/PSD/N/N) | | | | |
| √ | √ | | | | | | | | | | | 0 | 1 |
| | | | √ | | √ | | | | | | | 0 | 1 |
| | | | | | | | √ | | √ | | | 0 | 1 |
| *MT-ND3 gene* | | | | | | | | | | | | 2 | 2 |
| A10188G (N/B/N/N) | | G10365A (T-lc/B/N/N) | | A10398G (T-lc/B/N/N) | | | | | | | | | |
| | | √ | | √ | | | | | | | | 2 | 1 |
| √ | | | | √ | | | | | | | | 0 | 1 |
| *MT-ND4 gene* | | | | | | | | | | | | 1 | 0 |
| T11255C (T-lc/B/N/N) | | T11916A (D-lc/PRD/D/P) | | | | | | | | | | | |
| √ | | √ | | | | | | | | | | 1 | 0 |
| *MT-ND5 gene* | | | | | | | | | | | | 0 | 2 |
| A13966G (T-lc/B/N/N) | | A14128G (T-lc/PSD) | | | | | | | | | | | |
| √ | | √ | | | | | | | | | | 0 | 2 |
| *MT-CO1 gene* | | | | | | | | | | | | 0 | 1 |
| G6267A (D-lc/B/D/P) | | A7149G (T-lc/B/N/N) | | | | | | | | | | | |
| √ | | √ | | | | | | | | | | 0 | 1 |

Dark grey: possible risk, light grey: possible protective effect, white: presumably neutral missense variation combinations

B: Benign, Con: Controls, D: deleterious, lc: low confidence, N: neutral; P: pathogenic; Pat: Patients, PRD: probably damaging, PSD: possibly damaging, T: tolerated

occurred in the *MT-ATP6* (11) and MT-CYB (13) genes with a few such combinations manifesting in the *MT-ND2*, *MT-ND3*, *MT-ND4*, *MT-ND5* and *MT- CO1* genes. Some combinations occurred only among the patients, some only among the controls and the remainder occurred in both groups. Of the ten such combinations seen only among the patients, a unique combination of missense variants in the *MT-ATP6* gene and a unique combination of missense variants in the *MT-CYB* gene co-existed in two pre-menopausal obese patients. Thus the 10 unique combinations occurred only in 8 patients, of which 7 were obese and 4 were pre-menopausal. Three each had invasive lobular carcinoma and invasive ductal carcinoma (NOS/NST), 1 each had ductal carcinoma in-situ and metaplastic carcinoma. The proportion of obese patients carrying unique combinations of missense variants was significantly higher than the proportion of non-obese patients (21.2% vs 3.7%: P <0.05). However, the proportions carrying the unique combinations of missense variants that occurred only among the controls did not significantly differ between the obese and non-obese individuals.

## Rate of mutations across the coding region of the mt genome

Out of the 464 variants called through next generation sequencing 337 (72.629%) were in the coding region, whereas 127 (27.370%) were in the control region. The rate of mutation /1000 bp for the genes are shown in Table 3. *MT-tRNA* genes were excluded from this analysis as their short length may bias the output. The highest mutation rate/1000 bp was observed in the *MT-ATP6* gene in both patients and the controls among the total of 30 pairs analysed by NGS and in the patients in the subset of 20 pairs with invasive ductal carcinoma (NST/NOS). The highest mutation rate/1000bp for controls in the subset was observed in the *MT-ND2* gene. The second highest mutation rate was in the *MT-CYB* gene for the patients among the 30 pairs and in the *MT-ND2* gene for the controls. The second highest rate of mutations/1000 bp was

**Table 3. Mutation rates of different genes in the patients and the controls subjected to next generation sequencing.**

| Gene | | All histological types of breast cancer | | | | | Subset of invasive ductal carcinoma (NST/NOS) | | | | |
|---|---|---|---|---|---|---|---|---|---|---|---|
| | | Patients (n = 30) | | Controls (n = 30) | | Ratio (P/C) | Patients (n = 20) | | Controls (n = 20) | | Ratio(P/C) |
| Name | Length bp | Variant count | Variants per 1000 bp | Variant count | Variants Per 1000 bp | | Variant count | Variants per 1000 bp | Variant count | Variants per 1000 bp | |
| *MT-RNR1* | 954 | 6 | 6.289 | 13 | 13.626 | 0.46 | 5 | 5.241 | 13 | 13.626 | 0.38 |
| *MT—RNR2* | 1559 | 9 | 5.772 | 10 | 6.414 | 0.89 | 5 | 3.207 | 9 | 5.772 | 0.56 |
| *MT-ND1* | 956 | 9 | 9.414 | 18 | 18.828 | 0.5 | 4 | 4.184 | 15 | 15.69 | 0.27 |
| *MT–ND2* | 1042 | 19 | 18.234 | 24 | 23.032 | 0.79 | 14 | 13.435 | 19 | 18.234 | 0.73 |
| *MT-CO I* | 1542 | 18 | 11.673 | 17 | 11.024 | 1.05 | 10 | 6.485 | 16 | 10.376 | 0.63 |
| *MT–CO 2* | 684 | 8 | 11.695 | 10 | 14.619 | 0.8 | 8 | 11.695 | 5 | 7.309 | **1.6** |
| *MT–ATP8* | 207 | 1 | 4.83 | 4 | 19.323 | 0.25 | 0 | 0 | 2 | 9.661 | _ |
| *MT–ATP6* | 681 | 14 | 20.558 | 16 | 23.494 | 0.88 | 10 | 14.684 | 12 | 17.621 | 0.83 |
| *MT–CO 3* | 784 | 10 | 12.775 | 12 | 15.306 | 0.83 | 6 | 7.653 | 9 | 11.479 | 0.67 |
| *MT–ND3* | 346 | 5 | 14.45 | 4 | 11.56 | **1.25** | 4 | 11.56 | 3 | 8.67 | **1.33** |
| *MT-ND4L* | 297 | 3 | 10.101 | 4 | 13.468 | 0.75 | 2 | 6.734 | 2 | 6.734 | **1** |
| *MT -ND4* | 1378 | 17 | 12.336 | 25 | 18.142 | 0.68 | 12 | 8.708 | 19 | 13.788 | 0.63 |
| *MT- ND5* | 1812 | 27 | 14.9 | 33 | 18.211 | 0.82 | 18 | 9.933 | 28 | 15.452 | 0.64 |
| *MT–ND6* | 525 | 9 | 17.4 | 6 | 11.428 | **1.52** | 6 | 11.428 | 4 | 7.619 | **1.5** |
| *MT–CYB* | 1141 | 21 | 18.404 | 24 | 21.034 | 0.87 | 14 | 12.269 | 17 | 14.899 | 0.82 |

C: controls, P: patients

observed in *MT-ND2* gene and *MT-ATP6* for the patients and the controls respectively in the subset. *MT-ATP8* and *MT-RNR2* genes had the lowest mutation rate. *MT-ND3* and *MT-ND6* genes showed a higher mutation rate among the patients in all the groups and in addition patients in the subset showed a higher mutation rate in the *MT-CO2* gene when compared with the respective controls.

## Distribution of haplotypes and haplogroups

Among the 60 individuals subjected to NGS (i.e.: 30 patients and 30 controls), 56 haplotypes were observed. Three haplotypes were found in more than one individual (S6 Table) Majority were assigned to macro-haplogroup M (58.33%) whereas the remainder (41.66%) of the cohort were assigned to haplogroups descending from macro-haplogroup N. The distribution of haplogroups and the macro-haplogroups among the patients and controls are shown in Table 4 and S7 Table. Prevalence of different haplogroups did not significantly differ between the patients and the controls although certain haplogroups were seem only in one group. M65a@16311 (13.33%) was the commonest haplogroup and 4 patients and 4 controls were assigned to this.

## Further bioinformatic analyses of functional consequences of variations predicted to be pathogenic/deleterious

Among the variations predicted to be pathogenic/deleterious by several servers used, M31V variation in MT-ND1 has been reported in the Mitomap to be associated with Alzheimer's and Parkinson's diseases; other substitutions with diabetes mellitus and deafness (M32T and M31I) indicating functional consequences. Though SNAP2, SIFT and APOGEE predicted the M31V variation to be deleterious or pathogenic, observed frequency among the patients could not establish an association with breast cancer.

Despite conflicting reports on its association with Leber Hereditary Optic Neuropathy, direct functional consequences of G78S variation in *MT-CO3* gene is not evident. However the highly flexible G at position 78 is located at the interface with chain A facing an Asparagine residue which can act as an H bond acceptor or a donor (Fig 1).

Sequence alignment of the MT-ATP6 protein is shown in Fig 2. In the MT-ATP6 protein, at position 146, a T residue is conserved throughout all animal mtDNA. T146A substitution is located in close proximity to the E145, S148 and L156 where pathogenic mutations that alter

**Table 4. Distribution of macro-haplogroups and haplogroups among the 30 matched pairs subjected to NGS analysis.**

| Macro-haplogroup / Haplogroup | Patients | Controls | Total |
|---|---|---|---|
| | n (%) | n (%) | N (%) |
| Macrohaplogroup | | | |
| M | 17 (56.66%) | 18 (60%) | 35 (58.33%) |
| N | 13 (43.33%) | 12 (40%) | 25 (41.66%) |
| Haplogroup | | | |
| N | 1 | 0 | 1 |
| U | 7 | 9 | 16 |
| R | 3 | 2 | 5 |
| HV | 1 | | 1 |
| H | 1 | 1 | 2 |

Details are given in the S7 Table

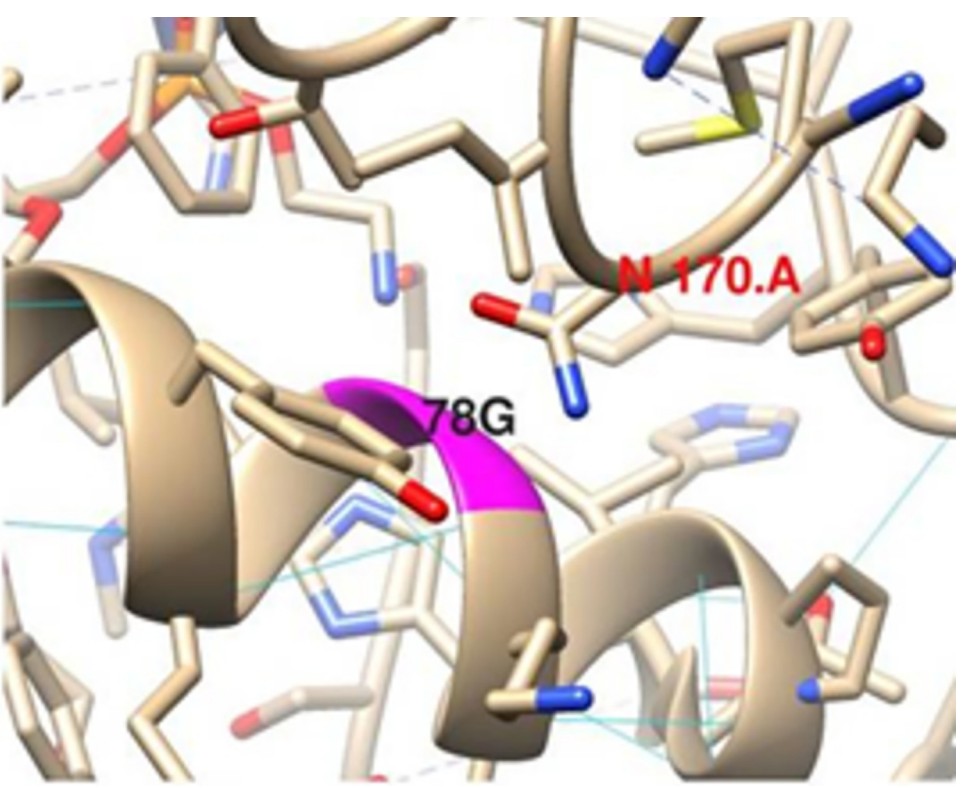

**Fig 1. Position of G 78 in MT-CO3.**

```
                                                              TMH 1
E. coli        MASENMTPQDYIGHHLNNLQLDLRTFSLVDPQNPPATFWTINIDSMFFSVVLGLLFLVLF   60
Yeast          ---MFNLLNTYITSPLDQ-------FEIRTLFGLQSSFIDLSCLNLTTFSLYTIIVLLVI   50
Human mutant   ---------------------------MNENLFASFIAPTILSLPATVLII-LFPPLL    30
Human wild-type ---------------------------MNENLFASFIAPTILGLPAAVLII-LFPPLL   30
                                                           .  ::*  .  .:   :    :.  ::

                                                              TMH 2
E. coli        RSVAKKATS----------GVPGKFQTAIELVIGFVNGSVKDMYHGKSKLIAPLALTIF   109
Yeast          TSLYTLTNNNNKIIGSRWLISQEAIYDTIMNMTKGQIGG-------KNWGLYFPMIFTLF  103
Human mutant   IPTSKYLINNRLITTQQWLIKLDSK----QMMATHNTKG-------RTWSLM---LVSLI   76
Human wild-type IPTSKYLINNRLITTQQWLIKLTSK----QMMTMHNTKG-------RTWSLM---LVSLI  76
                 .          .            .         :.  *        .  *    .:::

                         TMH 2                                    TMH 3
E. coli        VWVFLTALMLLFIDLDPYIAEHVLGLPALRVVPFADVNVTLSMALGVFILILFYSIKMK   169
Yeast          MFIFIANLISMIPYSFAL---------------SAHLVFIISLSIVIWLGNTILGLYKH   147
Human mutant   IFIATTNLLGLLPHSFTP---------------TEQLSMNLAMAIPLWAGAVIMGFRSK   120
Human wild-type IFIATTNLLGLLPHSFTP---------------TTQLSMNLAMAIPLWAGTVIMGFRSK  120
                 :::   **:.::* .:                :::.:  :.::  : .:  :   : .: :

                                                              TMH 4
E. coli        GIGGFTKELTLQPFNHWAFIPVNLILEGVSLLAKPVTIGLRLFANYAAELIFILIAGLL   229
Yeast          GWVFFS--LFVPAGTPLPLVPLLVIIETLSYFARAISLGLRLGSNILAGHLLMVILAGLT   205
Human mutant   IKNALA--HFLPQGTPTPLIPMLVIIEAISLLIQPMALAVRLTANITAGHLLMHLIGSAT   178
Human wild-type IKNALA--HFLPQGTPTPLIPMLVIIETISLLIQPMALAVRLTANITAGHLLMHLIGSAT  178
                 ::      :    . ::*: :*:* :* : : ::*.:** .*: **.*:: :..

                                                  TMH 5
E. coli        PWWSQWILNVPW--------------AIFHILFTAFIFMVLTIFVLSMASEEH   271
Yeast          FNFML--INLFTLVFGFVPLAMILAIMMLEFAIGIIQGYVWAILTASYLKDAVYLH   259
Human mutant   LDMST--INLPS--TFIIFTILILLTILEIAVALIQAYVFTLLVSLYLHDNT---   226
Human wild-type LAMST--INLPS---TLIIFTILILLTILEIAVALIQAYVFTLLVSLYLHDNT---  226
                 :*:         ::.: :  :*.:::  :*.  **
```

**Fig 2. Sequence alignment of MT-ATP-6 (subunit-a).** The primary sequence alignment of subunit-a from *Escherichia coli*, yeast, sequence carrying sequence variations described in this study (red) and human wild-type. The residues predicted to form transmembrane helices 3–5 (TMH3-5) are underlined. The residues shaded blue are important for proton movement. The highly conserved and essential R159 is shaded yellow.

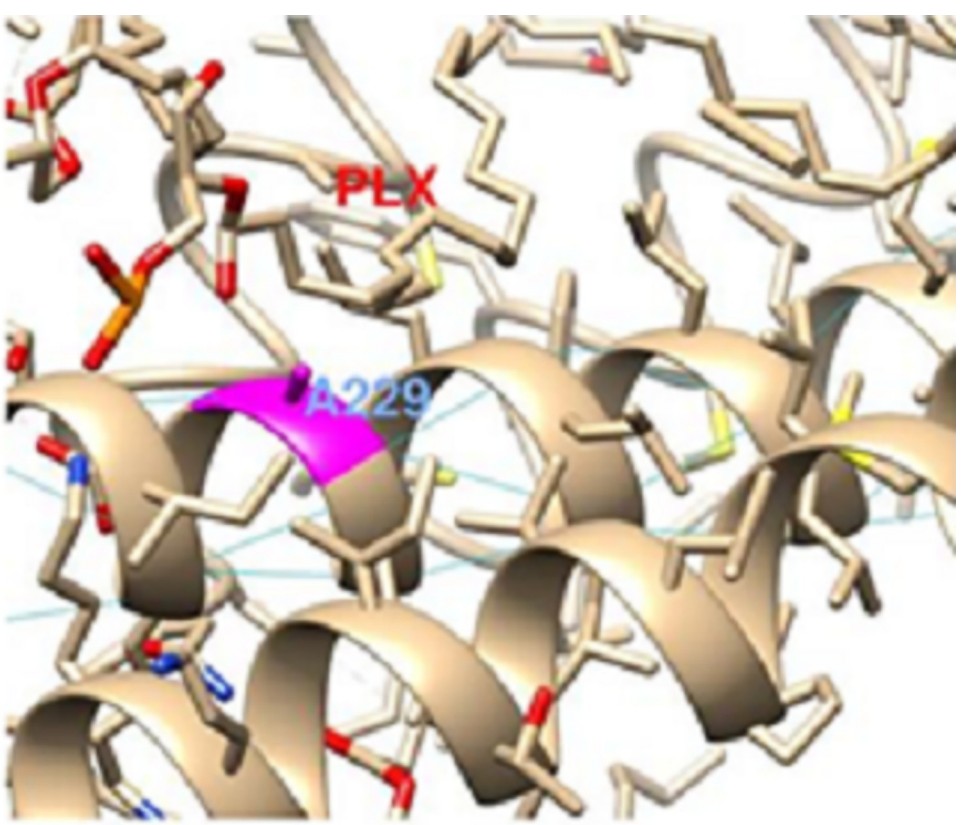

**Fig 3. Position of A 229 in MT-ATP6.**

the coupling efficiency of the ATP synthase have been reported [35, 55]. All the tools except APOGEE predicted the T146A variation to have a deleterious effect. Further the variation had a frequency <0.1% in GenBank sequences indicating an evolutionary pressure for selecting T at position 146.

In the *MT-CYB* gene product, A229T variation has been previously indicted in para crystalline inclusions with exercise intolerance [55] and obesity [56] suggesting functional consequences. A229 is located in a hydrophobic helix exposed to membrane lipids closer to their hydrophilic heads (Fig 3).

## Results of molecular dynamics simulation

Molecular dynamics and simulation (MD) studies were carried out in order to determine the stability and convergence of wild type human cytochrome C oxidase (PDB ID: 5Z62) and its mutant G78S as well as wild type mitochondrial ATPase and mutant T146A. MD simulation of both human cytochrome C oxidase wild type and mutant G78S, and wild type mitochondrial ATPase and mutant T146A appeared well embedded in POPE (1-Palmitoyl-2-oleoyl-sn-glycero-3-phosphoethanolamine) membrane (Fig 4A and 4B), however, the wildtype exhibited more closed conformation as compared to the mutant (Fig 4A).

Simulation of 20 ns displayed stable conformation while comparing RMSD values. The RMSD of Cα-backbone of the wild type human cytochrome C oxidase Chain A exhibited a deviation of 2.2 Å (Fig 5A). RMSD till 16 ns seemed to be well equilibrated; later a small peak hike is observed (Fig 5A). On the other hand a very stable converged RMSD of the trajectory

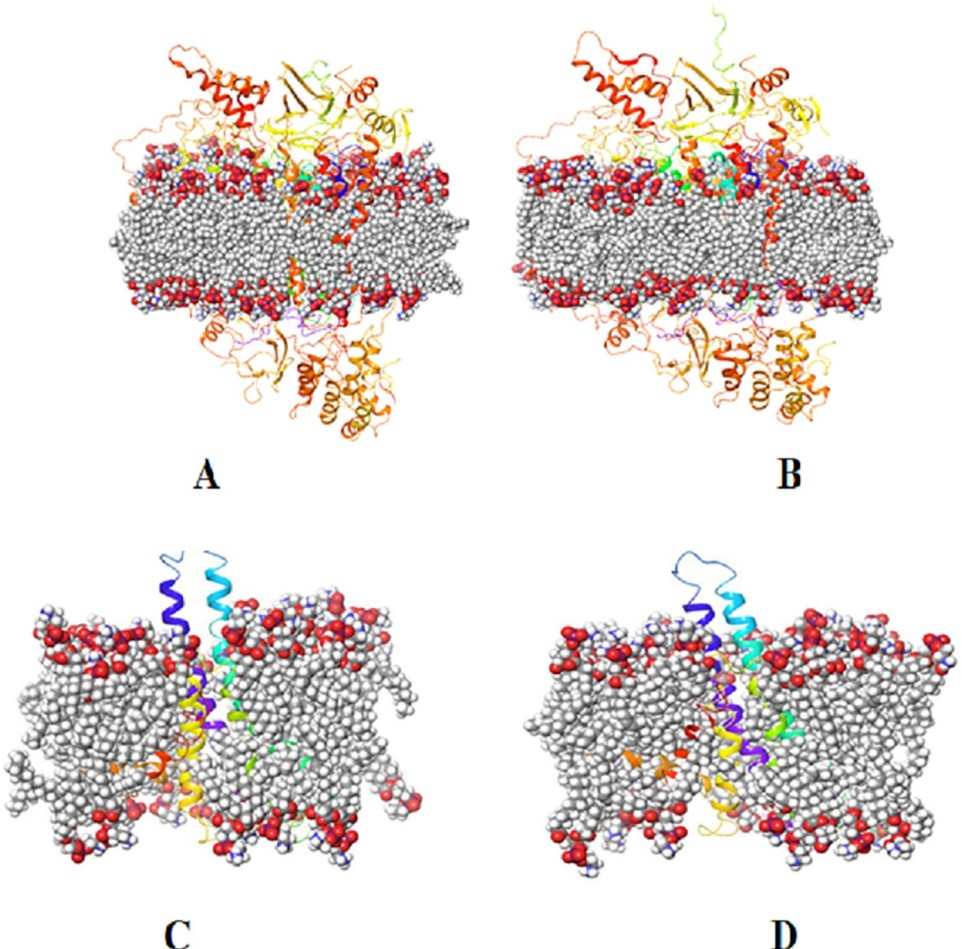

**Fig 4.** Conformations of 20 ns simulation of (A) wild type cytochrome C oxidase (B) G78S (C) wild type mitochondrial ATPase (D) T146A mutant proteins embedded in POPE membrane displaying closed and open conformations, respectively.

for G78S mutant was obsered with maximum deviation of 1.9 Å for chain A (Fig 5A). How-ever, the mutant T146A exhibited RMSD 3.8 Å and the wild type mitochondrial ATPase dis-played 4.5 Å (Fig 5B). Here in this study, for the radius of gyration wild type protein human cytochrome C oxidase Chain A Cα-backbone showed lowering of peak 22.95 to 22.80 Å and then almost stable throughout till 20 ns (Fig 5E). The mutant T146A showed significant lower-ing of peak from the beginning to 5 ns of simulation while the rest became stable (Fig 5F). On the other hand, the wild type exhibited stable radius of gyration 21.6 to 21.4 Å (Fig 5F). Hydro-gen bonds were seen in significant numbers between G78 with metal-complex throughout the simulation time 100 ns (Fig 5C). A consistent number of hydrogen bonds were observed between G78 with chain B (Average 2 numbers) and a similar pattern was oberved in the G78S mutant throught the simulation time (Fig 5D) that might facilitate to conform into a stable complex.

Principal component analysis (PCA) of the MD simulation trajectories for chain A of wild type protein and G78S mutant domain was analyzed to interpret the randomized, global motion of the atoms of amino acid residues. The internal coordinate's mobility into three-dimensional space in the spatial time of 20 ns was recorded in a covariance matrix. The

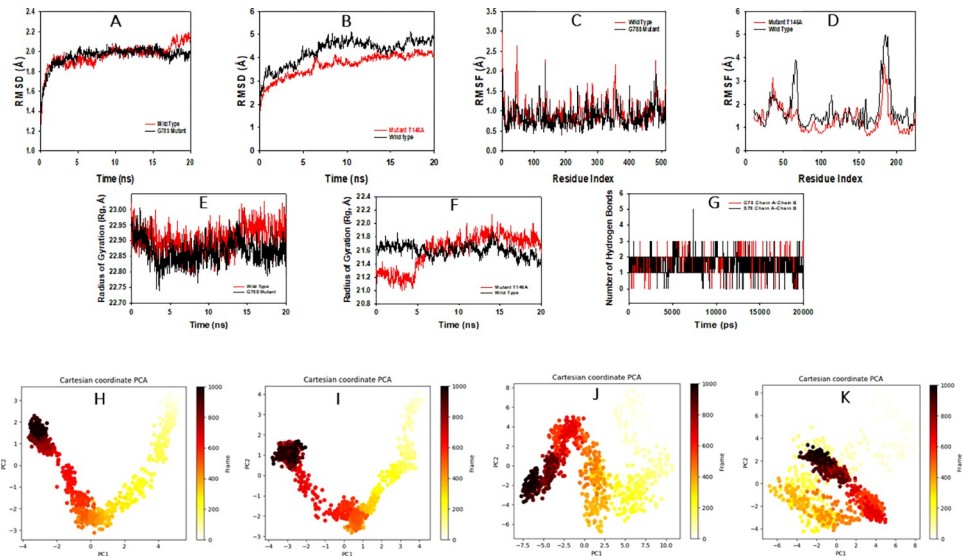

**Fig 5. Analysis of MD simulation trajectories of 20 ns time scale.** (A) RMSD plot displaying the molecular vibration of Cα backbone of wild type (red), and G78S mutant (black). (B) RMSD plot displaying the molecular vibration of Cα backbone of wild type mitochondrial ATPase (black), and T146A mutant (red). (C) RMSF plots show the fluctuations of respective amino acids throughout the simulation time 20 ns wild type (red), and G78S mutant (black). (D) RMSF plots show the fluctuations of respective amino acids throughout the simulation time 20 ns wild type mitochondrial ATPase (black), and T146A mutant (red). (E) The radius of gyration plots for the deduction of compactness of protein wild type (red), and G78S mutant (black). (F) Radius of gyration plots for wild type mitochondrial ATPase (black), and T146A mutant (red). (G) Number of hydrogen bonds formed between G78 and chain B (red) and S78 and chain B (black) during 20 ns simulation time scale. Principal Component Analysis (PCA) of (H) wild type chain A domain and (I) G78S mutant, (J) wild type ATPase and (K) mutant T146A, into unordered (yellow) to ordered (black) clusters, while comparing PC1 and PC2 for 20 ns simulation trajectories.

rational motion of each trajectory is interpreted in the form of orthogonal sets or Eigenvectors. MD simulation trajectory of Cα atoms of wild type protein displayed more unordered initial trajectories but later near to 20 ns clustered more toward positive Eigen values in PC1 and PC2 modes (Fig 5E, black) while more ordered movement of the frames observed in G78S mutant and oriented the last 50 frames (Fig 5F, black) into a cluster toward the center of the plot (more toward 0) while comparing PC1 and PC2 modes. Similarly, Eigen values in PC1 and PC2 modes appeared more unordered for the wild type mitochondrial ATPase (Fig 5J), but ordered for the mutant T146A (Fig 5K).

## Discussion

We carried out a comprehensive analysis of the coding region of the mt genome in sporadic breast cancer patients of Sinhalese ethnicity using next generation sequencing. In an additional cohort comprising of patients and controls having similar inclusion and exclusion criteria, selected areas of the mt genome were sequenced using Sanger sequencing. Mt genomic variants identified were compared with age, body mass index and menopausal status matched healthy controls in order to identify those significantly predisposing to or protecting from sporadic breast cancer.

As expected a large number variants were identified both in the patients and the controls and a few variants were highly prevalent in both groups. Majority of the variants occurred in the coding region and the present analysis was focused on these. Although for a few variants prevalence differed between the patients and the matched controls, the difference was not statistically significant. Thus no pathogenic variants common to patients were identified. When

we predicted the function of the protein coded by missense variants using SIFT and PolyPhen2 servers, a consistent deterioration of function/ possibly or probably damaging effect with both servers were observed only for 6 variants, and these occurred in either group or both groups. Interestingly all the missense variants among the patients predicted to be deleterious by SIFT or possibly/probably damaging by PolyPhen 2 were located in 4 genes, namely *MT-ATP8*, *MT-ATP6*, *MT-ND4* and *MT-CYB*. When the CADD and APOGEE Databases were examined additional variants were found to be designated as deleterious or pathogenic. Previous researchers detected only 24 mitochondrial variants predicted to be deleterious by SIFT and probably damaging by PolyPhen2 in a cohort of 436 French Nationals with familial breast cancer but testing negative for *BRCA1* and *BRCA2* [57]. Similar to our findings they did not observe any pathogenic variant common to the study cohort and reported the highest mutation rate in *MT-ATP* and *MT-CYB* genes.

Missense variants A8860G in the *MT-ATP6* gene and, C14766T and A15326G in the *MT-CYB* gene which had a prevalence of 93% or more among our patients and controls were seen at a 100% prevalence in a cohort of Malaysian breast cancer patients comprising of women of Chinese, Malay and Indian ethnicities [58]. A8860G has been reported as a risk factor for breast cancer in Chinese women [10]. Our findings preclude a role for this variant on its own as a risk factor for sporadic breast cancer among Sinhalese women. Neither SIFT nor Poly-Phen2 predicted A8860G to alter protein function and neither CADD nor APOGEE reported it to be deleterious or pathogenic, but SNAP2 score for this variant was 66 and the ConSurf score was 7. It has been reported that SNAP2 outperforms other tools for protein function prediction [41]. The occurrence of A8860G with one or more missense variants within the *MT-ATP6* gene appeared to be associated with the risk of (G8584A, T8701G, T8705C, T8843C, A8962G or G9064T) or protection from (G8572A, A8812G) breast cancer at an individual level in the present study. Of these coexisting variants, T8705C, A8962G and G8572A had a SNAP2 score >50 while the remainder had SNAP2 scores <50. Co-existence of A8860G with T8705C or with A8962G was seen in one patient each and co-existence with G8572A was seen only in one control.

Although only one missense variant in the *MT-CYB* gene was predicted to be possibly damaging by PolyPhen2 and none as deleterious by SIFT except with low confidence, several other variants observed in the present study had been reported to be deleterious by CADD and pathogenic by APOGEE. A15326G occurring with G14861A, as well as occurring with C14776T and one other variant (G15119A, G15497A, G15803A) appeared to increase the risk of breast cancer at an individual level. Similarly A15326G and C14776T in the presence of A14927G or G15314A and G15431A appeared to be protective. Of the variants seen in *MT-CYB*, A15326G, G15314A and G15497A resulted in SNAP2 scores >50. A15326G and G15314A coexisted in 5 patients and 3 controls whereas A15326G and G15497A coexisted in one patient.

Role of A10398G in the *MT-ND3* gene in relation to breast cancer had been inconsistent. This missense variant causes an amino acid substitution within the complex I, NADH dehydrogenase subunit (ND3), from threonine to alanine. The revised Cambridge reference sequence [27] refers to the A allele as the wild type, but the G allele at this locus defines the African haplogroups L1 and L3 and the M macro-haplogroup which is more prevalent in South Asia including Sri Lanka. Increased breast cancer risk was conferred by the G allele in European-American and Malay populations [13, 59], and by the A allele in African-American and North Indian women [11, 12]. In contrast, a study on South Indian women and a meta-analysis [14], and a recent study on a Malaysian cohort comprising of Chinese, Malay and Indian women [58] did not find an association of this locus with breast cancer. A synergistic interaction of A10398G with A12308G in the *MT-TL2* [13], and with T4216C in the *MT-ND1* [11] genes has been reported. In the present study although the A12308G in *MT-TL2* was

observed in some of the patients and the controls this did not co-exist with A10398G whereas T4216C in *MT-ND1* was not observed. The prevalence of the G allele of A10398G was 58.33% among our patients and the controls. Given the role of the G allele in defining macro-haplogroup M and the prevalence of this macro-haplogroup among Sri Lankan ethnicities [60], lack of an association of this locus with breast cancer in Sinhalese women from Sri Lanka is not surprising. Although previous authors have attributed both alleles to increased oxidative stress, and to apoptosis of normal cells, none of the bioinformatics tools used in the present study predicted a significant alteration of protein function due to this polymorphism.

G11719A in the *MT-ND4* gene is a synonymous variant and it occurred at a prevalence over 90% among our patients and the controls and at 100% prevalence in the Malaysian cohort [58]. It has been implicated in *BRCA* deficient breast cancer in an Italian cohort, but the authors also report a similar prevalence in healthy controls [61]. These authors reported the lowest prevalence of this variant in patients negative for *BRCA* mutations compared to *BRCA1* mutated and sporadic breast cancer patients. However, the numbers in each group were very small with approximately 10 subjects per group, hence their findings may be fortuitous.

While the variants that occurred only among the patients or the controls can be considered as potential risk or protective variants, their prevalence was very low to arrive at a solid conclusion. Similar to our observations, almost all the variants identified in breast cancer patients in the French National cohort [57] and the Malaysian cohort [58] are in fact very low prevalence variants. At an individual level, given the individual's genetic background, altered function of the protein may predispose to or protect one from carcinogenesis as evident from the unique combinations of missense variant within a gene seen exclusively among patients or controls. Unfortunately, the very low prevalence of variants that appear to alter protein function precludes these being developed as genetic biomarkers to predict sporadic breast cancer risk in Sri Lankan women of Sinhalese ethnicity.

However, in certain categories of women, mtDNA variants may play a considerable role in breast carcinogenesis. Missense and the pathogenic variants of mtDNA had a higher prevalence in pre-menopausal patients. Certain mtDNA variants when occurred in combinations appeared to predispose obese women to breast cancer.

With regard to the Molecular Dynamics Simulation results for the two missense variants G78S in human cytochrome C oxidase and T146A in human mitochondrial ATPase, there were interesting observations. Root mean square fluctuation measures the proteins conformation flexibility and stability. Here, the wild type protein human cytochrome C oxidase Chain A, exhibited significant fluctuations at residues 50, 360–370 and 490 which might be due to greater flexibility of the residue positions. In contrast, the G78S variant exhibited more rigid conformations having less flexible residues at the same positions. In mitochondrial ATPase, wild type showed more flexible residues at 55–70 and 17–190 in comparison to T146A variant. Radius of gyration (Rg) is the measure of compactness of the protein. Significant lowering and stable gyration (Rg) indicate highly compact orientation of the protein. G78S variant of cytochrome C oxidase exhibited more lowering of peaks than wild type indicating more compact structure of the G78S variant. Number of hydrogen bonds between the G78 in wild type protein and S78 in mutant protein with the nearest chain B suggests a significant interaction stability of the complex having G78 variant. The PCA analysis of both G78S variant form of human cytochrome C oxidase and the T146A variant form of human mitochondrial ATPase interprets more flexible scattered trajectories due to the protein structure's orientation into clusters. Therefore, it can be suggested that both G78S variant form of human cytochrome C oxidase and the T146A variant form of human mitochondrial ATPase are likely to be more stable than wild type complexes having optimal global motion.

Although we observed three shared haplotypes there were no known first or second degree relatives among the study subjects recruited. Those who shared haplotypes were from different geographical localities, but they may have shared a more recent maternal ancestor. Similar to previous observations on Sinhalese women from Sri Lanka [21] we failed to find a significant association of the mt haplogroups with breast cancer. In the present study we had haplogroup data only for the 30 matched pairs analysed using NGS. Although among a cohort of 60 matched pairs we previously found a weak association between the mitochondrial haplogroup M65a+@16311 and the sporadic breast cancer risk, in the current study we observed similar frequencies for this haplogroup among the patients and the controls. Interestingly one of our healthy controls was assigned to haplogroup U2e1a1, which has been observed in an ancient mt genome from Ukraine that belonged to the Late Eneolithic (3350–3200 BC) culture [62]. We also observed H6a1a haplogroup in one patient and one control. H6a1a is derived from H6a which has been assigned to an ancient mt genome (2400–2300 BC) from Poland associated with the Corded Ware culture [62]. U2e1a1 and H6a1a are seen in modern Western Eurasian ethnic groups more frequently than in the Indian subcontinent. Both these haplogroup had not been reported from Sri Lanka before. Observation of U2e1a1 and H6a1a among our study groups further supports previous observations of significant contribution of West Eurasian maternal lineages to certain Sri Lankan ethnic groups including Sinhalese [60].

Mitochondrial genome is a site of frequent variants and population specific variants allow its use in understanding events such as migration patterns and peopling of different geographies. Population specific variations in the mitochondrial genome are now well established. In such a background finding variants that modulate disease risk in one or several populations being a very common polymorphism or a rare variant in another population is not surprising. Our overall results suggest that genomic mtDNA variants are unlikely to become predictive biomarkers for sporadic breast cancer at least for Sinhalese women from Sri Lanka, except perhaps for specific categories such as pre-menopausal women and obese women. The possible role of somatic mtDNA variants in carcinogenesis in this population yet remain to be ascertained.

Using a case control study where breast cancer patients and healthy controls were matched for several confounding variables, we attempted to identify significant differences in the prevalence of coding region variants of the mt genome between the two groups. When compared with the previous studies in other populations, ours was a matched-pairs case control study which eliminated the effect of several confounding variables. Furthermore, it was limited to a single ethnic group to eliminate the effect of ethnicity on the mtDNA variants. We also stratified our analysis by menopausal status, tumour histology and body size to unravel any differences in the outcome and observed a greater association of mtDNA variants with breast cancer in pre-menopausal women and in obese women. Several *in-silico* tools were used for assessing functional significance of the variants and selected variants were subjected to Molecular Dynamic simulations. To our knowledge, many studies which previously reported association of mitochondrial variants with breast cancer did not assess functional significance of the variants and the few which did so used a fewer number of *in-silico* tools. Molecular Dynamic simulation of identified variants does not seem to have been carried out in previous studies.

## Conclusions

In view of the potential of using mtDNA variants to predict sporadic breast cancer, inconsistencies regarding association of specific mtDNA variants with breast cancer across populations and lack of such data for Sri Lankans, we analysed the mt genome in sporadic breast cancer of Sinhalese women and assessed the functional significance of the variants identified *in-silico*.

Mt genome in sporadic breast cancer patients and matched healthy controls showed a large number of variants including missense variants which were either polymorphisms or rare variants. However, variants with a significantly higher prevalence among patients that can be developed as a predictive biomarker were not evident. Unique combinations of missense variants that occurred exclusively among the patients in certain genes may have predisposed the carriers to carcinogenesis at an individual level while the opposite may have occurred with unique combinations of missense variants that occurred exclusively among the controls. The presence of a higher number of missense variants in premenopausal patients and unique combinations of missense variants in obese patients suggest a role for mtDNA in breast carcinogenesis in pre-menopause and obesity. A larger cohort needs to be studied to clarify the significance of mtDNA variants in these categories.

Molecular dynamics simulations of the G78S variant form of human cytochrome C oxidase and the T146A variant form of human mitochondrial ATPase indicated that both variants are comparatively more compact and stable within the complex which might positively or negatively affect their function. Further *in vitro* experiments are needed to evaluate the exact effect of these variants on their function.

## Supporting information

**S1 Fig. Flow chart of the study protocol.**
(TIF)

**S2 Fig. Coverage distribution along mitochondrial genome of all 60 samples in NGS, after removal of duplicates.**
(TIF)

**S1 Table. Body size of sporadic breast cancer patients and matched controls.**
(DOCX)

**S2 Table. Histological type of cancer in 60 sporadic breast cancer patients.**
(DOCX)

**S3 Table. Variants identified at >5% within either population (patients: N = 30; controls: N = 30) in the coding region of the mt genome by next generation sequencing excluding genomic regions *MT-ND3, MT-ND6, MT-ND4L, MT-CYB, MT-TT*.**
(DOCX)

**S4 Table. Variants identified at >5% within either population (patients: N = 60; controls: N = 60) in the genomic regions *MT-ND3, MT-ND6, MT-ND4L, MT-CYB, MT-TT* by next generation sequencing and Sanger sequencing.**
(DOCX)

**S5 Table. All the missense variations identified and the prediction of protein function and pathogenicity.**
(DOCX)

**S6 Table. Shared haplotypes.**
(DOCX)

**S7 Table. Detail description of the mt haplogroups in 30 matched pairs analysed using next generation sequencing.**
(DOCX)

**S8 Table. List of acronyms.**
(DOCX)

## Acknowledgments

We thank all the study participants, staff of the Apeksha Hospital and, Mr. Kanchana Senanayake and Ms. Anoma Jayasoma of the Institute of Biochemistry, Molecular Biology and Biotechnology, University of Colombo for IT and laboratory management respectively.

## Author Contributions

**Conceptualization:** Ruwandi Ranasinghe, Kamani H. Tennekoon.

**Data curation:** Lakshika P. Jayasekera, Joanne T. Kotelawala, Kanishka de Silva.

**Formal analysis:** Lakshika P. Jayasekera, Kanishka S. Senathilake.

**Funding acquisition:** Ruwandi Ranasinghe, Kamani H. Tennekoon.

**Investigation:** Lakshika P. Jayasekera, Kanishka S. Senathilake, Joanne T. Kotelawala, Kanishka de Silva, Priyanka H. Abeygunasekara, Renuka Goonesinghe, Kamani H. Tennekoon.

**Methodology:** Lakshika P. Jayasekera, Ruwandi Ranasinghe, Kanishka S. Senathilake, Kamani H. Tennekoon.

**Project administration:** Kamani H. Tennekoon.

**Resources:** Ruwandi Ranasinghe, Kanishka de Silva, Priyanka H. Abeygunasekara, Renuka Goonesinghe, Kamani H. Tennekoon.

**Supervision:** Ruwandi Ranasinghe, Kanishka de Silva, Kamani H. Tennekoon.

**Validation:** Lakshika P. Jayasekera, Ruwandi Ranasinghe, Kanishka S. Senathilake, Kamani H. Tennekoon.

**Visualization:** Lakshika P. Jayasekera, Ruwandi Ranasinghe, Kanishka S. Senathilake.

**Writing – original draft:** Lakshika P. Jayasekera, Kanishka S. Senathilake.

**Writing – review & editing:** Lakshika P. Jayasekera, Ruwandi Ranasinghe, Kanishka S. Senathilake, Joanne T. Kotelawala, Kanishka de Silva, Priyanka H. Abeygunasekara, Renuka Goonesinghe, Kamani H. Tennekoon.

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
