## [Decision Letter · Decision Letter 0]

22 Dec 2022

PONE-D-22-32207Mitochondrial genome in sporadic breast cancer: A case control study and a proteomic analysis in a Sinhalese cohort from Sri LankaPLOS ONE

Dear Dr. Tennekoon,

Thank you for submitting your manuscript to PLOS ONE. After careful consideration, we feel that it has merit but does not fully meet PLOS ONE’s publication criteria as it currently stands. Therefore, we invite you to submit a revised version of the manuscript that addresses the points raised during the review process.

We look forward to receiving your revised manuscript.

Kind regards,

Arabinda Ghosh

Academic Editor

PLOS ONE

“Financial support for this work was received from National Research Council (NRC), Sri Lanka (Grant No: NRC/17-020). We thank all the study participants, staff of the Apeksha Hospital and, Mr. Kanchana Senanayake and Ms. Anoma Jayasoma of the Institute of Biochemistry, Molecular Biology and Biotechnology, University of Colombo for IT and laboratory management respectively.”

“This work was financially supported by the National Research Council (NRC), Sri Lanka (Grant No: NRC/17-020). The funders had no role in the study design, data collection and analysis, decision to publish or preparation of the manuscript.”

Reviewers' comments:

Reviewer's Responses to Questions

**Comments to the Author**

1. Is the manuscript technically sound, and do the data support the conclusions?

Reviewer #1: Yes

Reviewer #2: Yes

2. Has the statistical analysis been performed appropriately and rigorously? 

Reviewer #1: Yes

Reviewer #2: Yes

3. Have the authors made all data underlying the findings in their manuscript fully available?

Reviewer #1: Yes

Reviewer #2: No

4. Is the manuscript presented in an intelligible fashion and written in standard English?

Reviewer #1: Yes

Reviewer #2: Yes

5. Review Comments to the Author

Reviewer #1: The manuscript is quite good and sounds scientifically good but I recommend some minor corrections before acceptance:

1. The abstract must be made more crisp for better understanding with a graphical abstract which shapes the whole work.

2. The concluding segment of introduction is lagging with some more information's and the need of this work with a flow diagram.

3. Authors must take care of typo and grammar errors.

4. Authors must add a new section which displays the new findings in contraction of the old system and their new findings

Reviewer #2: 1. In DNA extraction method, what does it mean by “..from the remainder”? Explain the remainder.

In MD simulation “….using an NVT ensemble for 100 ns” is quite long. Generally for 10-15 ns is enough. Recheck

2. What is the exact production run time in MD simulation?

3. Only 20 ns run is quite impossible to suggest stability of a mutant protein. In my opinion at least 200 ns or more simulation time will depict the clear picture.

4. Explain the background of the study in the abstract. Abstract is too brief.

5. Highlight the scope of the study in the conclusion section preferably in a single paragraph.

6. Indicate exclusively the objectives of the study in the last paragraph of the introduction. Concluding remarks should be removed. So delete the sentences "Although many germ line variants were

unique to breast cancer they were mostly confined to one or very few individuals. Some mutants

appeared to be more stable than their wild type counterparts".

7. The work is well-written, although there are several acronyms scattered throughout. It is encouraged that writers use complete words rather than abbreviations for greater clarity.

6. PLOS authors have the option to publish the peer review history of their article (what does this mean?). If published, this will include your full peer review and any attached files.

Reviewer #1: **Yes: **Nobendu Mukerjee

Reviewer #2: No

---

## [Author Response · Author response to Decision Letter 0]

26 Jan 2023

Editor’s comments Response

• A marked-up copy of your manuscript that highlights changes made to the original version. You should upload this as a separate file labeled 'Revised Manuscript with Track Changes'. DONE

 This is the letter being uploaded DONE

 DONE

“This work was financially supported by the National Research Council (NRC), Sri Lanka (Grant No: NRC/17-020). The funders had no role in the study design, data collection and analysis, decision to publish or preparation of the manuscript.”

Please include your amended statements within your cover letter; we will change the online submission form on your behalf. Reference to funding in acknowledgements was removed

Included in the cover letter that there is no change to the Funding Statement submitted via online system

While revising your submission, please upload your figure files to the Preflight Analysis and Conversion Engine (PACE) digital diagnostic tool, https://pacev2.apexcovantage.com/. PACE helps ensure that figures meet PLOS requirements. To use PACE, you must first register as a user. Registration is free. Then, login and navigate to the UPLOAD tab, where you will find detailed instructions on how to use the tool. If you encounter any issues or have any questions when using PACE, please email PLOS at figures@plos.org. Please note that Supporting Information files do not need this step. All Figures submitted have been passed by PACE

Reviewer 1 - 

The abstract must be made more crisp for better understanding with a graphical abstract which shapes the whole work. Abstract has been modified to include the background to the study as requested by reviewer 2.

Graphical abstract has been included.

The concluding segment of introduction is lagging with some more information's and the need of this work with a flow diagram. 

The concluding segment in the introduction has been deleted as per suggestions of reviewer 2 and the objectives have been inserted.

A flow diagram has been inserted as supplementary figure 1 and referred to in the text. 

Authors must take care of typo and grammar errors Corrected

Authors must add a new section which displays the new findings in contraction of the old system and their new findings Inserted "Using a case control study where breast cancer patients and healthy controls were matched for several confounding variables, we attempted to identify significant differences in the prevalence of coding region variants of the mt genome between the two groups. When compared with the previous studies in other populations, ours was a matched-pairs case control study which eliminated the effect of several confounding variables. Furthermore, it was limited to a single ethnic group to eliminate the effect of ethnicity on the mtDNA variants. We also stratified our analysis by menopausal status, tumour histology and body size to unravel any differences in the outcome and observed a greater association of mtDNA variants with breast cancer in pre-menopausal women and in obese women. Several in-silico tools were used for assessing functional significance of the variants and selected variants were subjected to Molecular Dynamic simulations. To our knowledge, many studies which previously reported association of mitochondrial variants with breast cancer did not assess functional significance of the variants and the few which did so used a fewer number of in-silico tools. Molecular Dynamic simulation of identified variants does not seem to have been carried out in previous studies." 

Reviewer 2 - comments 

 In DNA extraction method, what does it mean by “..from the remainder”? Explain the remainder.

In MD simulation “….using an NVT ensemble for 100 ns” is quite long. Generally for 10-15 ns is enough. Recheck Replaced “remainder” with “remaining 20 samples for clarity”.

It was run for 10ns, not 100ns. It was corrected in the manuscript 

What is the exact production run time in MD simulation? 20 ns

Only 20 ns run is quite impossible to suggest stability of a mutant protein. In my opinion at least 200 ns or more simulation time will depict the clear picture. In view of the practical difficulty of simulating large protein complexes with the membrane environment, Short 20 ns simulations were carried out to capture mainly the local motions and essential collective dynamics of the mutant and wild type proteins (Karami et al, 2018). Dynamic simulations of 20ns (Dorosh et al, 2013, Zhang et al, 2017, Mercer et al, 2018, Muneeswaran et al, 2018) or even less (Blinov et al, 2009) has been applied in previous studies for similar predictions and even to reproduce some experimental evidence of the stability of mutant proteins in silico (Zhang et al, 2017)

Added the phrase likely to be in the abstract (last line) and discussion (9th para) when referring to stability of the mutant proteins. 

Explain the background of the study in the abstract. Abstract is too brief. “Mitochondrial genome was analysed in sporadic breast cancer patients of Sinhalese ethnicity” has been replaced by 

“Breast cancer is the commonest malignancy in women and the majority occurs sporadically with no hereditary predisposition. However, sporadic breast cancer has been studied less well than the hereditary form and to date hardly any predictive biomarkers exist for the former. Furthermore, though mitochondrial DNA variants have been reported to be associated with breast cancer, findings have been inconsistent across populations. Thus we carried out a case control study on sporadic breast cancer patients and healthy controls of Sinhalese ethnicity (N=60 matched pairs) in order to characterize coding region variants associated with the disease and to identify any potential biomarkers. Mitochondrial genome was fully sequenced in 30 pairs and selected regions were sequenced in the remaining 30 pairs. Several in-silico tools were used to assess functional significance of the variants observed. A number of variants were identified among the patients and the controls.” 

 Highlight the scope of the study in the conclusion section preferably in a single paragraph. Inserted “In view of the potential of using mtDNA variants to predict sporadic breast cancer, inconsistencies regarding association of specific mtDNA variants with breast cancer across populations and lack of such data for Sri Lankans, we analysed the mt genome in sporadic breast cancer of Sinhalese women and assessed the functional significance of the variants identified in-silico.”

Indicate exclusively the objectives of the study in the last paragraph of the introduction. Concluding remarks should be removed. So delete the sentences "Although many germ line variants were

unique to breast cancer they were mostly confined to one or very few individuals. Some mutants

appeared to be more stable than their wild type counterparts". Deleted “Although many germ line variants were unique to breast cancer they were mostly confined to one or very few individuals. Some mutants appeared to be more stable than their wild type counterparts”

Inserted “Using a matched-pairs study design coding region of the mitochondrial genome was compared between sporadic breast cancer patients and healthy controls matched for age, body mass index and menopausal status. Prevalence of identified variants were compared between the two groups and subsets based on the histology of the tumour and body size. Rate of mutation was computed for each mitochondrial gene. Variants identified were tested in-silico for functional significance using several tools and selected variants were further analysed using Molecular Dynamic simulations.

The work is well-written, although there are several acronyms scattered throughout. It is encouraged that writers use complete words rather than abbreviations for greater clarity. Several acronyms which appeared three times or less were removed and full name used. Other acronyms which were kept are common acronyms such as DNA, NGS, PCR etc. or names of genes, tools, Databases. Two acronyms in relation to histopathological diagnosis was kept as they are within parenthesis as these are commonly used abbreviations in histopathology. A supplementary file giving all the acronyms has been included as S8 Table. 

References for comment 3 of Reviewer 2

Blinov N, Berjanskii M, Wishart DS, Stepanova M. Structural Domains and Main-Chain Flexibility in Prion Proteins. Biochemistry. 2009;48: 1488–1497. doi:10.1021/bi802043h

Dorosh L, Kharenko OA, Rajagopalan N, Loewen MC, Stepanova M. Molecular Mechanisms in the Activation of Abscisic Acid Receptor PYR1. PLoS Comput Biol. 2013;9: e1003114. doi:10.1371/journal.pcbi.1003114

Karami Y, Bitard-Feildel T, Laine E, Carbone A. “Infostery” analysis of short molecular dynamics simulations identifies highly sensitive residues and predicts deleterious mutations. Scientific Reports 2018;8: 16126. doi:10.1038/s41598-018-34508-2

Mercer RCC, Daude N, Dorosh L, Fu Z-L, Mays CE, Gapeshina H, et al. A novel Gerstmann-Sträussler-Scheinker disease mutation defines a precursor for amyloidogenic 8 kDa PrP fragments and reveals N-terminal structural changes shared by other GSS alleles. PLOS Pathogens. 2018;14: e1006826. doi:10.1371/journal.ppat.1006826

Muneeswaran G, Pandiaraj M, Kartheeswaran S, Sankaralingam M, Muthukumar K, Karunakaran C. Molecular dynamics simulation approach to explore atomistic molecular mechanism of peroxidase activity of apoptotic cytochrome c mutants. Informatics in Medicine Unlocked. 2018;11: 51–60. doi:10.1016/j.imu.2018.04.003

Zhang D, Lazim R. Application of conventional molecular dynamics simulation in evaluating the stability of apomyoglobin in urea solution. Scientific Reports. 2017;7: 44651. doi:10.1038/srep44651

---

## [Decision Letter · Decision Letter 1]

30 Jan 2023

Mitochondrial genome in sporadic breast cancer: A case control study and a proteomic analysis in a Sinhalese cohort from Sri Lanka

PONE-D-22-32207R1

Dear Dr. Kamani Hemamala Tennekoon

We’re pleased to inform you that your manuscript has been judged scientifically suitable for publication and will be formally accepted for publication once it meets all outstanding technical requirements.

Kind regards,

Arabinda Ghosh

Academic Editor

PLOS ONE

Reviewers' comments:

Reviewer's Responses to Questions

**Comments to the Author**

1. If the authors have adequately addressed your comments raised in a previous round of review and you feel that this manuscript is now acceptable for publication, you may indicate that here to bypass the “Comments to the Author” section, enter your conflict of interest statement in the “Confidential to Editor” section, and submit your "Accept" recommendation.

Reviewer #1: All comments have been addressed

Reviewer #2: All comments have been addressed

2. Is the manuscript technically sound, and do the data support the conclusions?

Reviewer #1: Yes

Reviewer #2: Yes

3. Has the statistical analysis been performed appropriately and rigorously? 

Reviewer #1: Yes

Reviewer #2: Yes

4. Have the authors made all data underlying the findings in their manuscript fully available?

Reviewer #1: Yes

Reviewer #2: Yes

5. Is the manuscript presented in an intelligible fashion and written in standard English?

Reviewer #1: Yes

Reviewer #2: Yes

6. Review Comments to the Author

Reviewer #1: Thanks for the good revision of the manuscript. The paper could be accepted for publication. Best wishes.

Reviewer #2: All the comments given, were addressed properly. Therefore, the article is recommended to be published.

7. PLOS authors have the option to publish the peer review history of their article (what does this mean?). If published, this will include your full peer review and any attached files.

Reviewer #1: **Yes: **Nobendu Mukerjee

Reviewer #2: No

---

## [Editor Report · Acceptance letter]

1 Feb 2023

PONE-D-22-32207R1 

Mitochondrial genome in sporadic breast cancer: A case control study and a proteomic analysis in a Sinhalese cohort from Sri Lanka 

Dear Dr. Tennekoon:

I'm pleased to inform you that your manuscript has been deemed suitable for publication in PLOS ONE. Congratulations! Your manuscript is now with our production department. 

Kind regards, 

on behalf of

Dr. Arabinda Ghosh 

Academic Editor

PLOS ONE